# Spike-phase coupling patterns reveal laminar identity in primate cortex

Zachary W Davis[1]*[†], Nicholas M Dotson[1], Tom P Franken[1,2], Lyle Muller[3,4]*[†], John H Reynolds[1]*[†]

[1]The Salk Institute for Biological Studies, La Jolla, United States; [2]Department of Neuroscience, Washington University in St. Louis School of Medicine, St. Louis, United States; [3]Department of Mathematics, Western University, London, Canada; [4]Brain and Mind Institute, Western University, London, Canada

**Abstract** The cortical column is one of the fundamental computational circuits in the brain. In order to understand the role neurons in different layers of this circuit play in cortical function it is necessary to identify the boundaries that separate the laminar compartments. While histological approaches can reveal ground truth they are not a practical means of identifying cortical layers in vivo. The gold standard for identifying laminar compartments in electrophysiological recordings is current-source density (CSD) analysis. However, laminar CSD analysis requires averaging across reliably evoked responses that target the input layer in cortex, which may be difficult to generate in less well-studied cortical regions. Further, the analysis can be susceptible to noise on individual channels resulting in errors in assigning laminar boundaries. Here, we have analyzed linear array recordings in multiple cortical areas in both the common marmoset and the rhesus macaque. We describe a pattern of laminar spike–field phase relationships that reliably identifies the transition between input and deep layers in cortical recordings from multiple cortical areas in two different non-human primate species. This measure corresponds well to estimates of the location of the input layer using CSDs, but does not require averaging or specific evoked activity. Laminar identity can be estimated rapidly with as little as a minute of ongoing data and is invariant to many experimental parameters. This method may serve to validate CSD measurements that might otherwise be unreliable or to estimate laminar boundaries when other methods are not practical.

**\*For correspondence:**
zdavis@salk.edu (ZWD);
lmuller2@uwo.ca (LM);
reynolds@salk.edu (JHR)

[†]Co-senior authors

**Competing interest:** The authors declare that no competing interests exist.

## Editor's evaluation

Authors demonstrate powerful methods that can be applied across species to find reliable markers that characterize activity in different cortical layers. Authors provide compelling evidence for these methods that enable systematic comparisons between slow extracellular voltage fluctuations and spiking across cortical columns. The results are timely since linear multielectrode array recordings have become a state-of-the-art technique.

## Introduction

Linear array electrodes have become a ubiquitous electrophysiological tool for understanding the functional roles of neural populations across the layers of the cortex, their interactions, and the computations they perform. This understanding requires reliable assignment of neurons to their respective laminar compartments. Precise localization of individual neurons can be obtained by electrolytic lesion to mark the position of an electrode; however, this procedure is not practical in experiments where the same animal is used in multiple experimental sessions as is nearly always the case in experiments involving non-human primates. The gold standard for identifying laminar compartments from

functional recordings is current-source density (CSD) analysis (*Mitzdorf, 1985*; *Mitzdorf and Singer, 1978*; *Schroeder et al., 1998*). The CSD represents the second spatial derivative of local field potential (LFP) activity averaged to repeated events. These events are often sensory-evoked stimulation as in flashed visual stimuli in visual cortex (*Maier et al., 2010*; *Schroeder et al., 1991*; *Wang et al., 2020*) or short duration sounds in auditory cortex (*Happel et al., 2010*; *Lakatos et al., 2007*; *Szymanski et al., 2011*) which produce feedforward activation of the canonical cortical columnar circuit via activation of the input layer (*Gratiy et al., 2011*; *Mitzdorf, 1985*; *Mitzdorf and Singer, 1978*). Laminar compartments can then be assigned from the timing of subsequent patterns of current sources and sinks across electrode contacts on the linear array (*Mitzdorf, 1985*). The earliest current sink reflects the feedforward activation of the input layers (*Mitzdorf, 1987*; *Mitzdorf, 1985*), and the current sources above and below are used to estimate the boundaries with respect to superficial (layers 1–3) and deep (layers 5 and 6) cortical layers (*Self et al., 2013*). CSD analysis has been validated histologically (*Mitzdorf and Singer, 1979*; *Schroeder et al., 1991*; *Takeuchi et al., 2011*) and reliably captures known laminar differences in the functional properties of cortical layers, such as differences in evoked latencies across layers in response to driving input (*Bode-Greuel et al., 1987*; *Einevoll et al., 2013*; *Plomp et al., 2017*).

There are some limitations on the use of CSD for identifying cortical layers in electrophysiology recordings. CSD analysis has largely been limited to primary sensory areas where the types of sensory stimuli that can robustly and reliably generate the evoked responses necessary for averaging CSDs are well established. Although CSD analysis has been used in higher-order cortical areas where sensory-evoked responses are apparent, such as visual responses in frontal cortex (*Bastos et al., 2018*; *Godlove et al., 2014*), it is less clear what to use as a triggering event in non-sensory cortical areas to reveal similar laminar patterns. Another limitation of CSD analysis is that depth estimates can only be taken as the average across the period of sensory response collection, making it difficult to track electrode drift during a recording. Further, noise in recordings due to bad channels, variability in filtering parameters, or ambiguity in the identifying source–sink pattern could potentially lead to the elimination of otherwise useful data or inaccurate laminar assignment. Without an alternative means of estimating the depth of electrode contacts, analyses of cortical circuit function are at risk of using biased definitions of laminar identity resulting in spurious conclusions about layer-dependent neuronal features.

Here, we present a new method for determining laminar boundaries in cortical recordings based on spike-phase coupling patterns across linear array electrodes. Previous work has shown ongoing spiking activity is strongly coupled to the phase of LFP fluctuations in the cortex (*Davis et al., 2022*; *Destexhe et al., 1999*; *Dotson et al., 2014*; *Eeckman and Freeman, 1990*; *Esghaei et al., 2018*; *Haegens et al., 2011*) and these spike–field relationships can be diagnostic of circuit interactions in cortical systems (*Safavi et al., 2023*). We hypothesized that, as phase shifts occur across the layers of the cortex, the preferred phase angle of this spike–LFP relationship should be influenced by or reflect changes in laminar circuitry that contribute to differences in sources and sinks. We find that there is a phase reversal in spike–field coupling that reliably corresponds to the laminar boundary between input and deep layers as estimated by CSD. We therefore propose a novel methodology, called laminar-phase coupling (LPC), for identifying the laminar boundary between input and deep cortical layers. This pattern reliably reversed in awake recordings from both common marmosets and rhesus macaque in multiple cortical areas including extrastriate visual and prefrontal cortex. This method can be applied to the same data used for CSD analysis, but can also be applied to any cortical linear array recording data without the need for specific sensory stimuli as with CSD analysis. Data recorded under a variety of conditions, such as during behavioral tasks, spontaneous activity during fixation, or continuous data recorded in the dark all reliably reveal the same pattern of laminar-phase separation. LPC is largely invariant to filtering or referencing, and can be done on single- or multi-unit activity. The analysis does not require averaging and can be calculated online to estimate recording depth throughout the duration of an experiment. The preferred phase angle in the spike–field relationship is a simple yet effective measure for estimating contact depth in linear array recordings with respect to laminar boundaries and may help inform the study of cortical circuits in situations where CSD measurements are ambiguous or where CSD methods are impractical or ineffective.

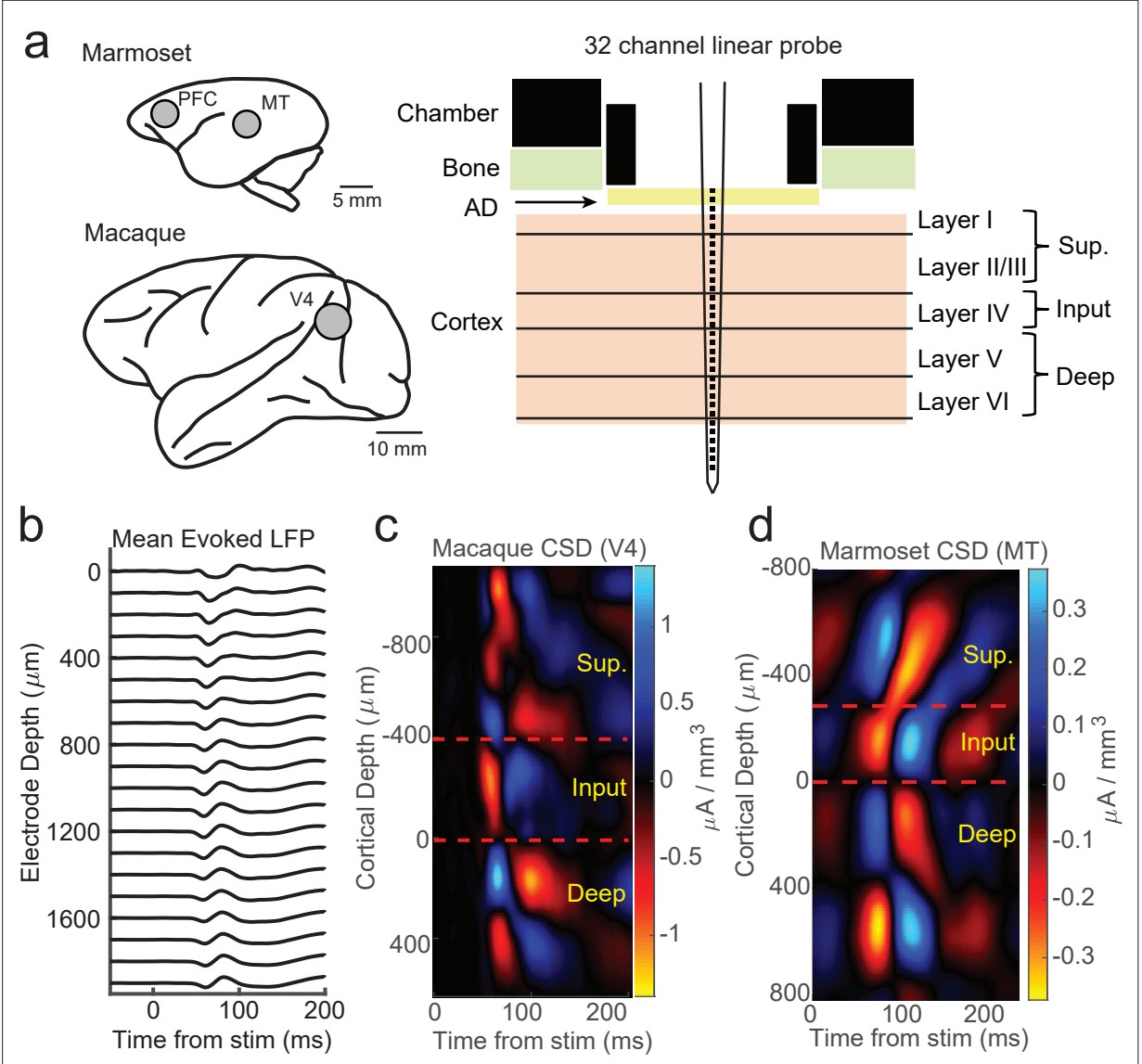

**Figure 1.** Current-source density (CSD) analysis reveals laminar boundaries in cortical recordings. (**a**) Schematic of recording locations in two marmosets (PFC or MT) and two macaques (V4). Recordings were made with a 32-channel linear silicon probe in an acute recording chamber penetrating through a silicone artificial dura (AD). (**b**) Average evoked local field potential (LFP) responses to a stimulus flash (N = 141 trials) in an example macaque recording session in area V4. Each line is the response measured on a single contact. The depth is in absolute distance from the most proximal contact. (**c**) CSD measurement from the example recording session in b. The input layer is defined as the bottom and top of the earliest current sink (red), with the superficial and deep layers defined as above and below the input layer, respectively. Depth is measured relative to the bottom of the input layer. (**d**) Same as c, but in an example marmoset MT recording session (N = 225 trials).

The online version of this article includes the following figure supplement(s) for figure 1:

**Figure supplement 1.** Power spectra of local field potential (LFP) recorded during current-source density (CSD) mapping.

**Figure supplement 2.** Comparison between Generalized Phase (GP) and Hilbert Transform for broad (5–50 Hz) and narrow (8–14 Hz) filtered signals.

## Results

We recorded electrophysiology data from 32-channel linear probes (Atlas Neuroengineering) inserted perpendicular to the cortical surface into cortical regions V4 in two macaque monkeys and middle temporal (MT) or pre-frontal cortex (PFC) in two marmoset monkeys performing head-fixed visual tasks under various experimental conditions (*Figure 1a*). CSD analyses were performed using stimulus-locked LFP epochs (*Figure 1b*). These epochs were locked to stimulus flashes in the case of V4 and MT recordings or full field flashes in the case of PFC recordings. As standard in CSD analysis (*Franken and*

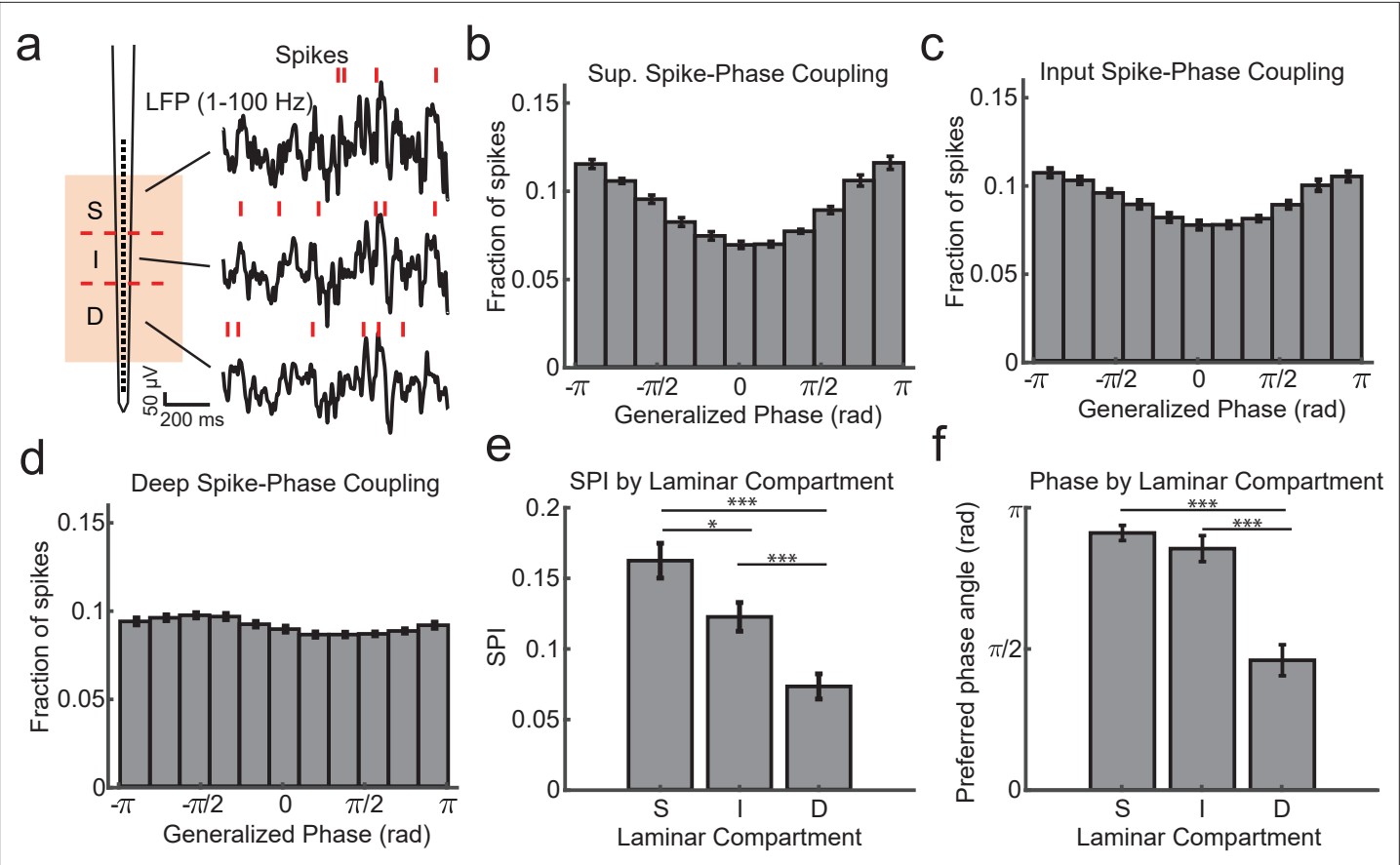

**Figure 2.** Within channel spike–local field potential (LFP) phase coupling strength and preferred phase angle varies across layers. (**a**) Spike–LFP phase coupling was measured by taking the phase of the LFP on a contact at the times when multi-unit spikes were detected on the same contact. (**b**) Spike-phase distribution for superficial contacts averaged across all recordings sessions ($N$ = 34 sessions across 4 monkeys; error bars denote standard error of the mean [SEM]). The spike-phase distribution was strongly peaked toward $\pm\pi$ rad. (**c, d**) Same as b, but for contacts in the input and deep layers. (**e**) The average spike-phase index (SPI) was significantly weaker for input and deep layers relative to the superficial layer (0.16 vs. 0.12 and 0.07; p = 0.017 S. v. I. and p < 1 × 10$^{-7}$ S. v D.; Wilcoxon signed-rank test; *p < 0.05, ***p < 0.0001). (**f**) There was no difference in the preferred LFP phase angle between superficial and input layers, but a significant difference between these layers and deep layers (−2.86 and −2.68 vs. −1.44 rad; F = 38.52, p < 1 × 10$^{-7}$ and F = 21.15, p = 0.00002, respectively; Watson–Williams test).

*Reynolds, 2021*; *Nandy et al., 2017*; *Schroeder et al., 1998*; *Self et al., 2013*; *Westerberg et al., 2022*), we took from the resulting evoked source–sink patterns the earliest sink (red) as the location of the input layers and assigned our reference depth relative to the input layer as the bottom of the current sink, with the boundaries to the sources (blue) above and below identifying the boundaries of the superficial and deep cortical layers (*Figure 1c, d*).

In order to test the relationship between spiking and LFP phase across the layers of the cortex, we first identified multi-unit spike times and measured the *generalized phase* (GP) of the LFP filtered from 5 to 50 Hz (*Davis et al., 2022*; *Davis et al., 2020a*). The use of a wider filter than traditionally used (i.e. 4–8 or 8–15 Hz) helps reduce phase distortions that occur when applying narrowband filters to signals that have broad spectral content (*Figure 1—figure supplement 1*). GP provides a measure of wideband phase that corrects for errors that may arise from applying the Hilbert Transform to signals with broader spectral content (*Figure 1—figure supplement 2*), although our results did not depend on our specific use of these techniques. We binned electrodes into superficial, input, and deep layers depending on their location relative to the source–sink pattern defined in the CSD analysis.

We first calculated the degree of spike coupling on each channel to the LFP phase measured on the same channel grouped by cortical layer (*Figure 2a*). This was done by taking the length of the circular resultant of the spike-phase distribution, which we call the spike-phase index (SPI). This index value ranged from 0 (perfectly uniform spike-phase distribution) to 1 (all spikes occur at a single

phase). The superficial channels had an average SPI value of 0.162 ± 0.012 (mean ± standard error of the mean [SEM]; $N$ = 34 sessions; *Figure 2b*). This was significantly greater than the input layer (SPI = 0.122 ± 0.010 mean ± SEM; p = 0.0003, Wilcoxon signed-rank test; *Figure 2c*), and both were significantly greater than the deep layers (SPI = 0.073 ± 0.009 SEM; p = 0.000003 S. v. D. and p = 0.000009 I. v. D.; *Figure 2d, e*). The preferred LFP phase angle (i.e. the circular mean of the phase distribution) for spiking in the superficial and input layers was 2.86 and −2.68 rad, respectively, which corresponds toward the trough of ongoing LFP fluctuations. This was significantly different from the preferred phase angle for deep layer spiking (−1.44 rad; $F$ = 38.52, $p < 1 \times 10^{-7}$ and $F$ = 21.15, p = 0.00002, respectively; Watson–Williams test; *Figure 2f*) suggesting that indeed the deeper layers may be distinguished by a change in the preferred spike–LFP phase angle relative to the superficial electrodes and this could be read out from the spike–LFP relationship across the depth of the cortex.

While the previous analysis examined the spike-phase relationship on each channel as a function of cortical depth on average, because of variability in the preferred mean phase on each channel (due to

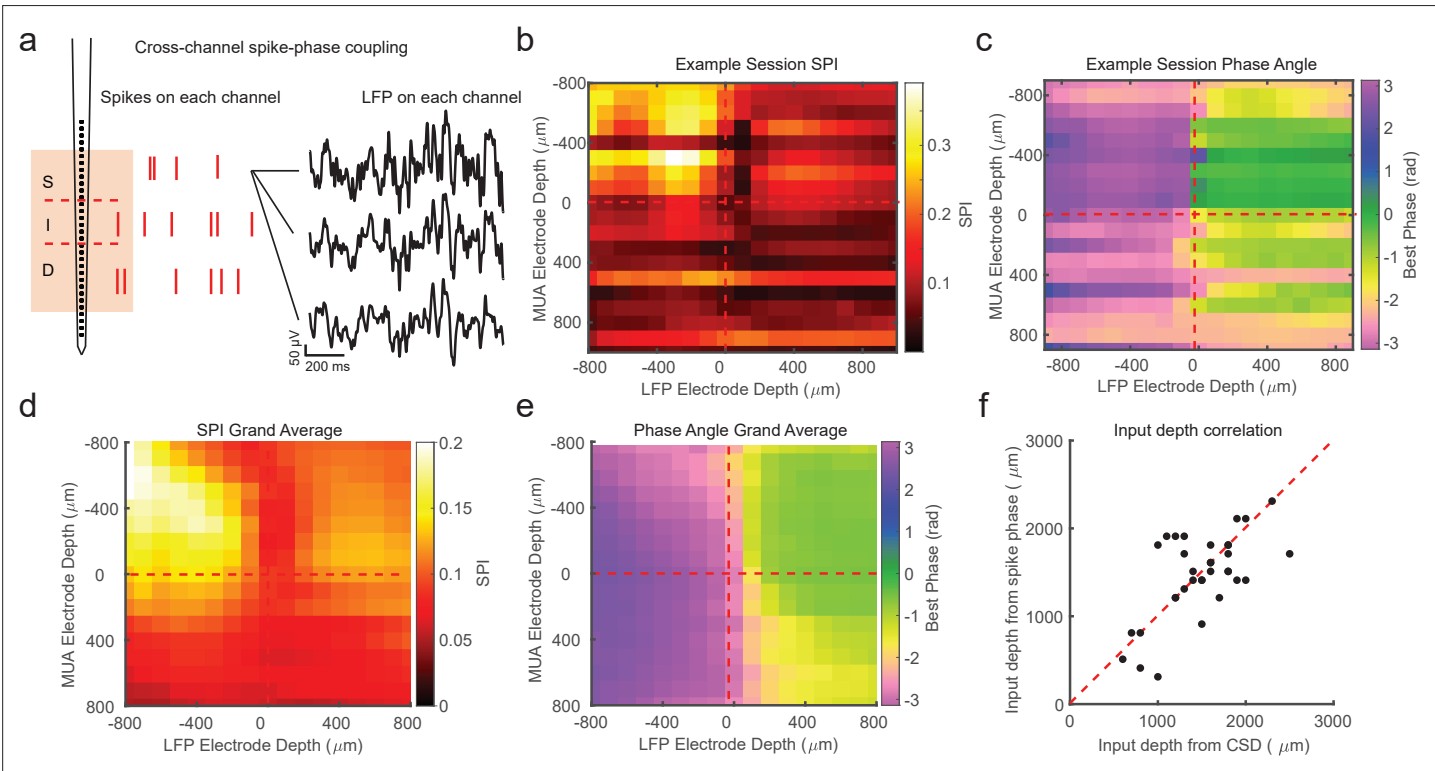

**Figure 3.** Cross-channel spike–local field potential (LFP) phase coupling reversal correlates with putative input layer. (**a**) Schematic displaying cross-channel phase coupling analysis. The spike times on each channel are compared against the phase of the LFP on each other channel, yielding a matrix of spike-phase coupling. (**b**) Example recording session cross-channel spike-phase index (SPI) values from marmoset MT. Red dashed lines indicate the estimated depth of the bottom of the input layer from current-source density (CSD) analysis. (**c**) Preferred phase angles for the cross-channel spike-phase relationships in b. Spikes across all channels preferred ±π rad phase angles from LFP measure on superficial and input electrodes, but preferred 0 rad phase angles from LFP measured on deep phase angles. The LFP phase reversal aligns well to the estimate of the input layer from CSD analysis (red dashed line). (**d**) Grand average cross-channel SPI across all recording sessions in MT, V4, and PFC aligned to the putative input layer from CSD analysis ($N$ = 34 sessions). (**e**) Grand average of the preferred phase angle for the data in d. (**f**) Scatter plot shows a significant correlation between the depth of the bottom of the input layer estimated from CSD analysis (x-axis) and the depth of the phase reversal (y-axis; Pearson's r = 0.64, p = 0.00005).

The online version of this article includes the following figure supplement(s) for figure 3:

**Figure supplement 1.** Laminar spike-phase reversal consistent across cortical areas.

**Figure supplement 2.** Laminar spike-phase pattern does not depend on filtering.

**Figure supplement 3.** Phase reversal is apparent when using well-isolated single units instead of MUA.

**Figure supplement 4.** Phase relationship does not depend on task conditions.

**Figure supplement 5.** Phase relationship is not dependent on referencing.

**Figure supplement 6.** Cross-channel laminar-phase coupling (LPC) can be estimated over short periods.

factors such as the strength of spike–LFP coupling and the number of spikes recorded), there was no significant correlation between the preferred phase angle and the depth of the electrode as measured with CSD (circular–linear correlation $r$ = 0.28; p = 0.24). A more sensitive measure is to relate the preferred phase angle in the spike–LFP relationship across the depths of all channels in the cortex. This was achieved by calculating the SPI from the multi-unit spiking activity on a given channel relative to the LFP phase angle measured on each contact across the depth of the recording (*Figure 3a*). We call the resulting matrix the cross-channel LPC. As an example, the first matrix row is derived from the spikes recorded on the first channel compared against the phase measured from the LFP on each of the 32 channels. The second row is derived from the spikes on the second channel compared against the phase measured from the LFP on each channel, and so on. The result is a 32-by-32 matrix where each cell represents the relation between the spikes on one channel and the LFP phase on another channel. However, not all of these channels are in the cortex, so we realigned the matrix based on estimates of cortical depth from the bottom of the input layer as identified from CSD analysis.

We first looked at the SPI values across the cross-channel LPC matrix. *Figure 3b* shows an example from a single recording (note that the diagonal in this matrix, from top left to bottom right, represents the approach in *Figure 2*). In this recording, we found that superficial spiking activity (defined based on CSD analysis) was strongly coupled to the LFP phase recorded on other superficial channels, which dropped off sharply below the input layer, and recovered when computed relative to the LFP phase on deeper channels. We next looked at the preferred phase angle for spiking activity on each channel relative to the LFP phase measured for different laminar compartments (*Figure 3c*). We found that spiking activity on all channels, regardless of cortical depth, tended to spike during ±π radian LFP phase angles measured on superficial channels. This phase relationship flipped such that spiking on all channels, regardless of cortical depth, tended to spike during 0 radian phase angles for LFP measured on deep channels. This phase reversal occurred about the channel we identified from CSD analysis as the boundary between the input and deep layers in the cortical column.

In order to see if this pattern held across our recordings, we then aligned all of our recordings ($N$ = 34) relative to the boundary that separated input and deep layers (from each recording's CSD) and computed grand averages for SPI (*Figure 3d*) and preferred phase angle (*Figure 3e*). The average pattern across recordings showed the laminar-phase reversal was well aligned to the estimate of the input layer from the CSDs across recordings. This was also true when we broke out the recordings to only average across sessions in each cortical area (MT, V4, PFC; *Figure 3—figure supplement 1*). To quantify how well the cross-channel LPC and CSD aligned on each individual recording session, we compared the depth of the laminar boundary estimated from CSD analysis to the depth at which the preferred spiking phase angle reversed (*Figure 3f*). There was a significant correlation between the depth estimated from the CSD and the phase reversal ($r$ = 0.64, p = 0.00005; Pearson's correlation) and no significant difference between the estimated depth values (mean difference = 271 ± 49 μm SEM; p = 0.98, Wilcoxon signed-rank test).

We next asked whether the observed phase relationship was parameter dependent. We first asked whether or not the specific use of a 5–50 Hz filter or the calculation of GP was necessary to see the laminar-phase reversal. To test this we filtered the data in a variety of commonly used frequency bands (4–8, 8–14, 15–30, and 30–50 Hz) and calculated phase using the Hilbert Transform (*Figure 3—figure supplement 2*). We found that the observed phase reversal was apparent in each case, indicating the results were not dependent on the filter or the method used to compute phase. We also tested whether the results depended on the use of multi-unit spiking activity. We performed the same analysis aligning cross-channel LPC from single units across recording sessions based on CSD depth (*Figure 3—figure supplement 3*). We observed the same relationship between laminar depth and preferred phase angle as when using multi-unit spiking activity. We also separated out different conditions during the same recording session where the electrode placement was unchanged (*Figure 3—figure supplement 4*). These included fixation during a visual task, freely viewing natural scene images, freely viewing in total darkness, and fixation during receptive field mapping. The cross-channel LPC revealed qualitatively similar depths for the laminar-phase reversal across experimental conditions within the same penetration. Finally, we explored the impact of referencing on the presence of the phase reversal. In our recordings, the electrode probe had a reference contact at the base of the shaft. In order to test whether our results depended on the location of the reference contact we re-referenced the LFP data to the shallowest contact, the deepest contact, or a common average

reference (*Figure 3—figure supplement 5*). The phase reversal persisted under all referencing conditions. These results indicate the phase relationship is a robust feature of columnar recordings and not dependent on a particular set of parameters.

One of the advantages of the LPC method is it can be done without the need for a triggering event, like a stimulus onset. Further, as little as a minute of continuously recorded data is sufficient to recover the phase reversal across channels. We find estimates can be derived from as few as hundreds of multi-unit spikes on each channel, although estimates are more reliable when spikes number in the thousands. As a result, depth estimates can be sampled at arbitrary points during recording sessions making this technique sensitive to tracking putative electrode drift over the course of a recording. To demonstrate this, we calculated LPC on sequential 3-min epochs during the first 15 min of an example recording session (*Figure 3—figure supplement 6*). We found the depth of the phase reversal moved over the course of 15 min. When comparing the depth after the first or last 2 min across all recordings, we found a significant difference in the estimated depth of the input layer from the spike-phase reversal (p = 0.005, Wilcoxon signed-rank test) such that the estimate of input layer depth consistently drifted deeper across recording sessions by an average of 132 ± 26 µm (SEM). This change in the depth estimate across the recording is consistent with electrode drift from movement of the probe relative to the cortex as the tissue settles at the start of the recording.

The cross-channel LPC relies on both spikes and LFP, but if a similar phase pattern can be observed in the LFP alone, it may not require spiking activity to detect the laminar boundary. To test this we measured the circular–circular cross-correlation between LFP phase on each channel and LFP phase on each other channel. On many recording sessions we found a characteristic phase correlation pattern that seemed to align well with both CSD and LPC depth estimates (*Figure 4a–c*), and across many recordings the phase correlations within layers could dissociate laminar compartments. However, this was not always the case. Some sessions yielded phase correlation patterns that were noisy and difficult to interpret, although knowing where the boundary was made it easier to identify the characteristic pattern (*Figure 4d, e*). While, on average, LFP phase correlations can be informative (*Figure 4f*), the spike–LFP phase relationship appears to provide a more robust measure of contact depth.

Across our recordings there were some instances (14/34) where there was 200 µm or more disagreement between CSD and LPC estimates for the location of the input layer relative to the probe contacts. *Figure 5a* shows an example CSD that has the characteristic source–sink pattern used to identify the input layer. This estimate was well aligned to the cross-channel LPC pattern previously described (*Figure 5b, c*). However, *Figure 5d* shows an example of a CSD profile with a similar source–sink pattern that is not well aligned to the cross-channel LPC pattern, which shows the characteristic phase reversal 600 µm below the CSD estimate (*Figure 5d, e*). In order to determine which measurement was more accurate in identifying cortical depth in these cases of disagreement, we sought to use each measure to replicate two findings from the literature that varied with cortical depth.

First, previous work in multiple cortical areas has shown that the superficial layers have lower firing rates as compared to the input and deep layers (*de Kock and Sakmann, 2009*; *Haegens et al., 2015*; *Lakatos et al., 2005*; *Leszczyński et al., 2020*). We identified recording sessions where the CSD and LPC estimate of input depth differed by more than 200 µm (*N* = 14 sessions) and we measured the average firing rate as a function of depth relative to either the CSD or LPC estimate of the input layer (*Figure 6a*). We found that on these sessions, the CSD aligned firing rates failed to recapitulate the finding of higher firing rates in the input and deeper layers (p = 0.09, Wilcoxon signed-rank test). Conversely, we did find significantly stronger input firing rates when aligned to the LPC estimate (p = 0.0003, Wilcoxon signed-rank test), suggesting the LPC estimate on these sessions was better aligned than the CSD to the ground truth.

We next examined a second previously reported finding that varied with laminar depth. Previous groups have reported an inversion in the spectral content of the LFP across layers, with the superficial layers exhibiting more high-frequency power (e.g. 80–200 Hz) and the deep layers exhibiting more low-frequency power (8–20 Hz) (*Maier et al., 2011*; *Maier et al., 2010*). The crossover between the power in higher and lower frequencies was reported to occur around the input layers defined from CSD measurements (*Bastos et al., 2018*), and validated histologically in multiple cortical areas (*Mendoza-Halliday et al., 2022*). To test whether we could identify a similar relationship in these ambiguous recordings, and whether that relationship was stronger when aligned to CSD or LPC estimates of depth, we calculated low- and high-frequency power on each channel in each session and

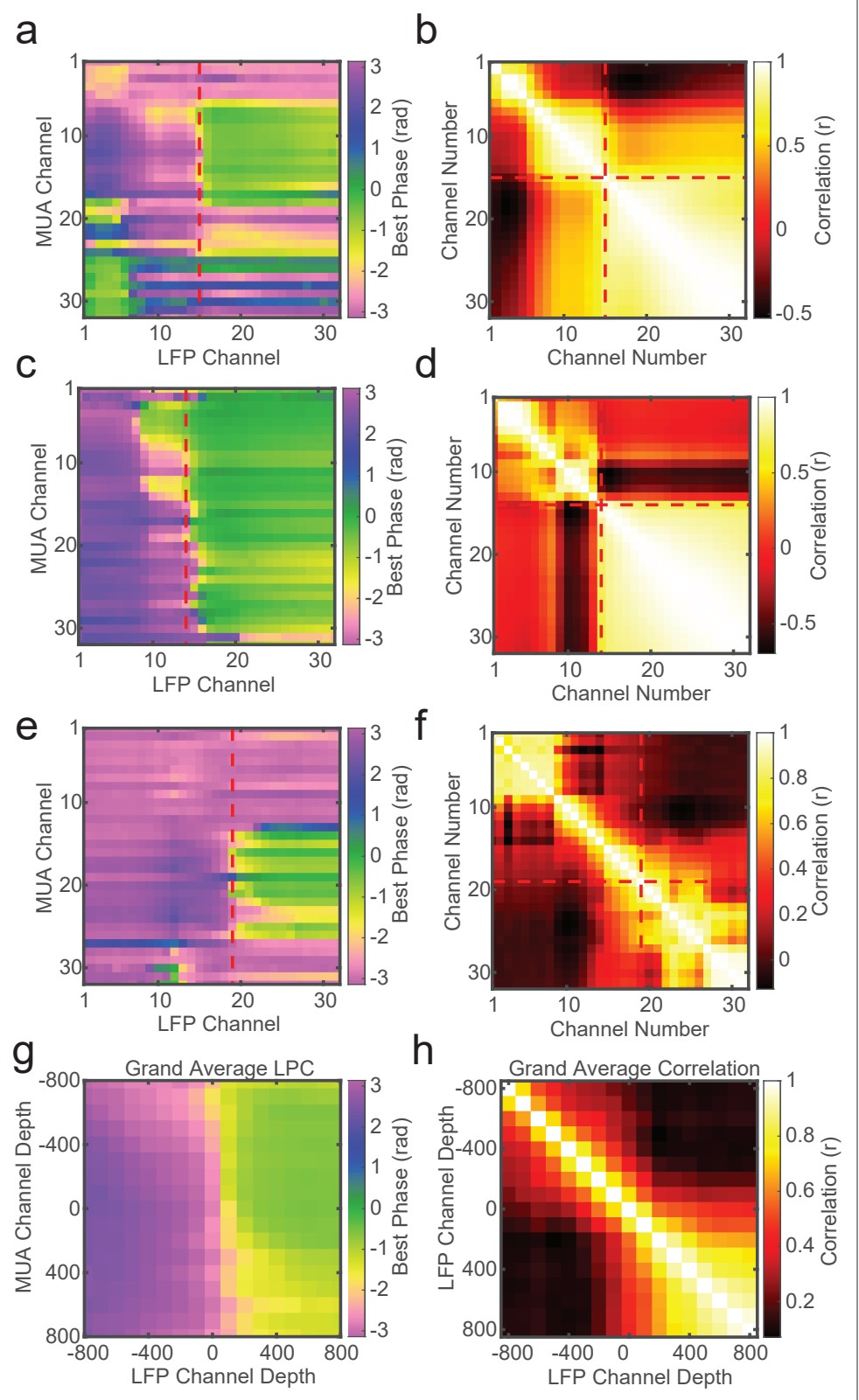

**Figure 4.** Local field potential (LFP)–LFP phase correlations can also reveal laminar boundaries, but are less consistent. (**a**) Preferred phase angle across all 32 channels from an example marmoset recording. The red dashed line indicates the current-source density (CSD) estimate of the bottom of the input layer. (**b**) Circular LFP phase correlation across channels. There is a strong within compartment correlation that aligns with the boundary

*Figure 4 continued on next page*

*Figure 4 continued*

between channels above and below the input/deep boundary (red dashed lines). (**c, d**) Same as panels a and b, but for a macaque recording. (**e, f**) Same as above, but an example of poor phase correlation within compartments despite strong phase reversal alignment with CSD. (**g, h**) Grand averages for preferred phase angle and LFP phase correlations across all recording sessions (*N* = 34). Plots were aligned to the putative input/deep layer boundary identified by CSD analysis.

---

aligned them to each depth estimate (*Figure 6b, c*). The depth estimate from the LPC measure was significantly correlated with the high/low power inversion (Pearson's *r* = 0.87, p = 0.00005) whereas the CSD estimate was more weakly, although still significantly correlated (Pearson's *r* = 0.65, p = 0.012), indicating that on the recording sessions where there was some disparity between the CSD and LPC estimate (*Figure 6d*), the LPC better captured the power inversion than the CSD measure (*Figure 6e, f*). These results suggest that when noise or error in assigning laminar compartments based on CSD analysis occur, the LPC phase reversal is a more reliable measure of laminar depth.

## Discussion

The cortical column is one of the fundamental computational circuits in the brain. In order to understand the function of disparate cortical areas and the contribution various cell types play in processing information through the columnar circuit, it is often necessary to identify and segregate the responses of neuronal populations based on their position in the layers of the cortex. Traditionally, for electrophysiological measurements, this has been achieved using CSD analysis of sensory-evoked responses. However, CSD analysis requires averaging across repeated discrete sensory events that may be difficult to generate in less well-studied cortical areas where the response selectivity is not apparent. While CSDs can also be calculated from intrinsic events such as up/down state transitions (*Senzai et al., 2019*) or bursts of oscillatory power (*Bollimunta et al., 2008*), it is less clear how reliable their patterns reflect the laminar organization of the underlying circuitry with respect to distinct anatomical boundaries – although methods have been proposed to recover spatial information from spontaneous activity (*Chand and Dhamala, 2014*). Further, because CSDs are measured across electrodes, a single noisy channel corrupts the contribution of the channel above and below potentially leading to an errant assignment of cortical layer boundaries. As histological approaches to recover individual recording tracts can be impractical, particularly in experiments in non-human primate where multiple recordings in the same area are the norm, it can be difficult to validate CSD measures or recover laminar information.

Here, we describe an alternative estimate of laminar boundary, LPC, that can supplement or in some cases replace CSD analysis. Using laminar boundaries defined by CSD, we find that spike-phase coupling inverts at the boundary between the input and deep layers of the cortex. This reversal provides a reliable and robust measure of the depth of linear array electrodes in cortex across a variety of analysis parameters and experimental conditions. Using LPC to identify laminar compartments, we are able to replicate the pattern of LFP power reversing across the input layer observed in multiple cortical areas and has been validated histologically (*Bastos et al., 2018*; *Maier et al., 2010*; *Mendoza-Halliday et al., 2022*). This measure is robust to different LFP filter settings with either single- and multi-unit data and can be applied to any arbitrary recording epoch so long as there are sufficient spikes across electrodes. Failing that, patterns of phase–phase correlation across channels may also help identify laminar boundaries. The ability to identify cortical depth across linear electrodes quickly and robustly permits the online identification of electrode positions relative to cortical layers and the tracking of electrode drift as the cortex settles following electrode penetration.

While the number of spikes necessary to recover the spike-phase inversion can be counted in the hundreds, the LPC measurement is more reliable in epochs in which thousands of spikes have occurred. It would be convenient if LFP phase alone were sufficient to identify cortical layers as this would eliminate the requirement for measuring multi-unit spiking on multiple electrodes, and indeed we find there are occasions where correlations in LFP phase are sufficient to provide strong evidence of the location of laminar boundaries. This is consistent with previous reports of dissociable patterns of LFP coherence between superficial and deep domains that show particular separation at the bottom of the input layer (*Maier et al., 2010*). However, we find the reliability of this pattern of LFP phase

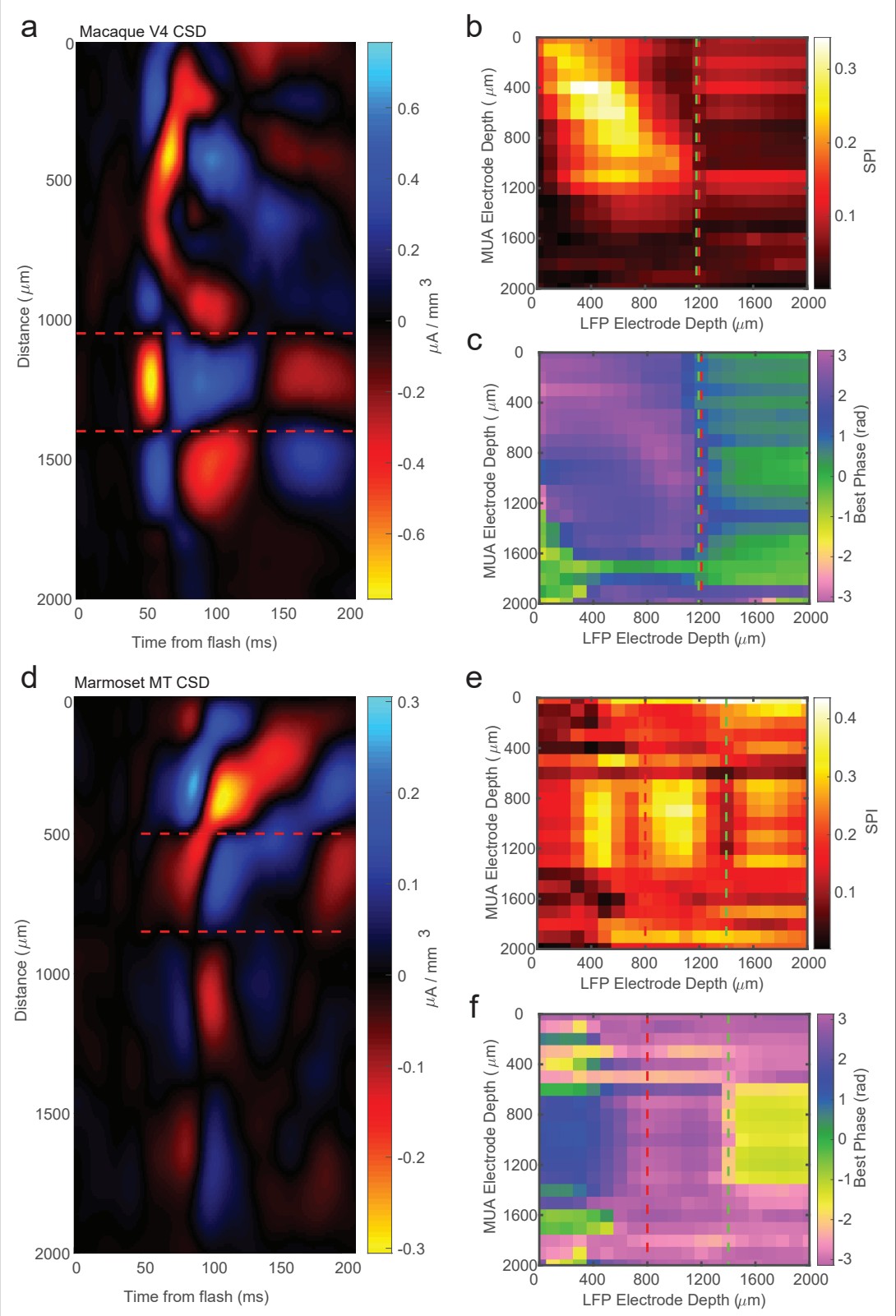

**Figure 5.** Current-source density (CSD) analysis is not always consistent with laminar-phase reversal. (**a**) Example CSD from macaque V4 with estimate of input layer boundaries indicated by red dashed lines (*N* = 81 trials). (**b**) Cross-channel spike-phase index (SPI) for the same example recording as the CSD in a. Red line is the CSD estimate of the bottom of the input layer. Green line is the estimate from the phase reversal. (**c**) Cross-channel spike-phase pattern for same recording as in a and b. The phase reversal is well aligned to the CSD estimate of the input layer. (**d**) CSD from a different example

*Figure 5 continued on next page*

Figure 5 continued
recording session in marmoset MT (*N* = 661 trials). Laminar depth estimated the same as in a. (**e, f**) Cross-channel SPI and phase angle as in b and c. The CSD and phase reversal estimates do not align.

correlation varies across the recordings in our dataset, although the reason why is unclear. We do find measures of cortical depth are improved when LFP phase is combined with the preferred timing of spiking activity across the cortical layers. However, if multi-unit data is only weakly apparent in a linear array recording, depth information might still be recovered from LFP phase relationships across channels without requiring CSD analysis.

One interesting observation is that the superficial and input layers of cortex show stronger spike–LFP coupling than the deeper layers. Why might this be the case? It may be that the LFP is largely reflecting the shared synaptic inputs in the numerous connections in these cortical layers (*Buzsáki et al., 2012*). This is consistent with measurements of correlated variability in V2 where spike-count correlations were observed to be stronger in the superficial and input layers and weaker in the deep layers (*Smith et al., 2013*), or in V4 where the strongest correlations were observed in the input layers relative to the superficial or deep layers (*Nandy et al., 2017*). A different pattern has been observed

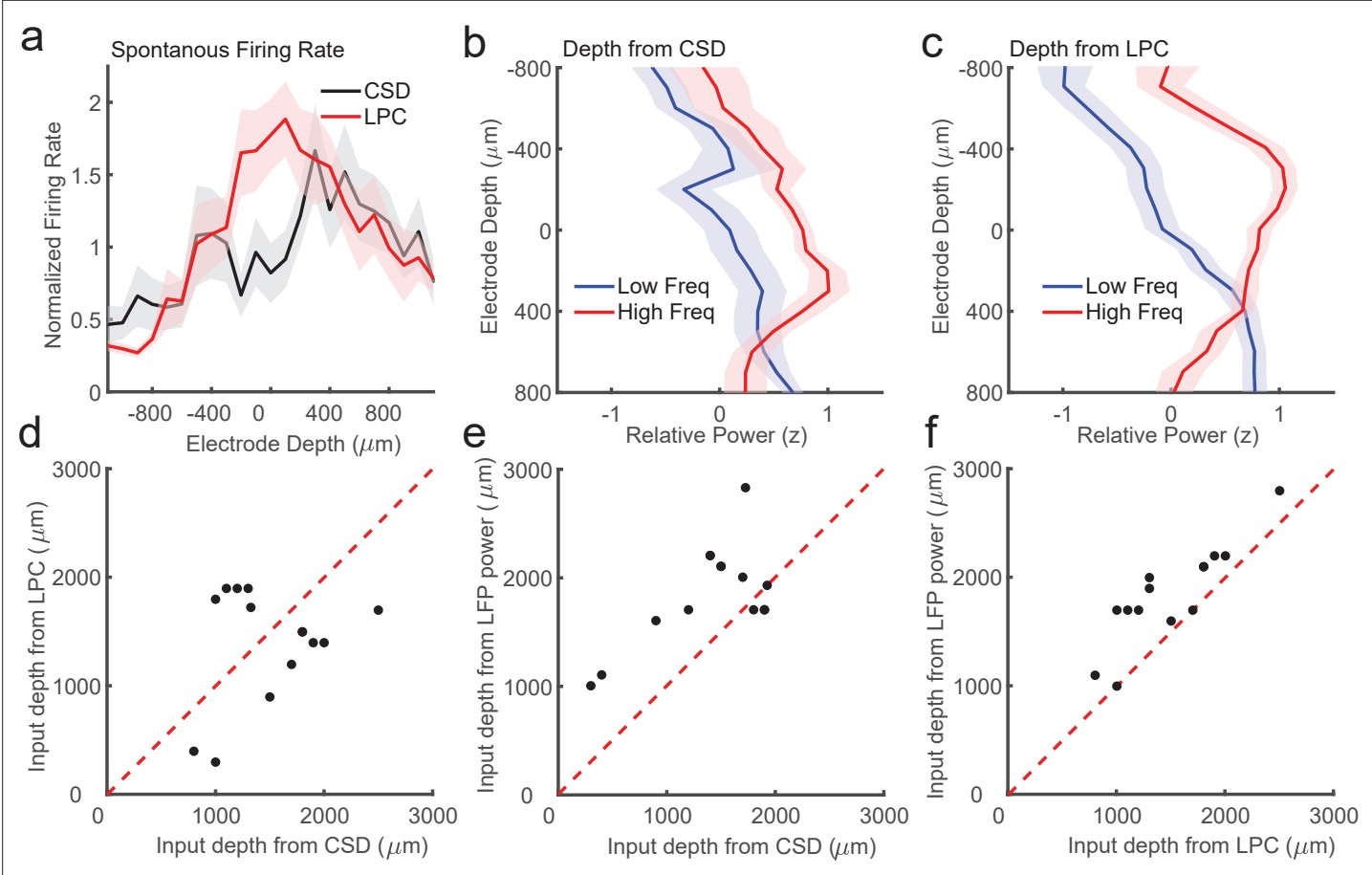

**Figure 6.** Laminar-phase coupling (LPC) pattern better replicates spike rate and local field potential (LFP) power dissociations than current-source density (CSD) when the two measures disagree. (**a**) Normalized spontaneous firing rates across sessions when CSD and LPC disagreed as a function of contact depth by more than 200 μm (*N* = 14; shaded regions denote standard error of the mean [SEM]). LPC captures the expected higher spontaneous firing rates around the input layer. (**b**) Mean *z*-scored LFP power across sessions in lower frequencies (8–30 Hz, blue line) and higher frequencies (65–100 Hz, red line) as a function of electrode depth referenced to the CSD estimate for sessions. (**c**) Same as in a, but referenced to the depth estimate from the LPC phase reversal across sessions. The laminar power relationship is more pronounced when using the phase reversal instead of CSD estimate. (**d**) Scatter plot showing the disagreement in the depth of the input layer estimated from CSD (*x*-axis) and the depth from LPC (*y*-axis) in this subset of recording sessions. (**e**) Scatter plot showing the alignment of the CSD input depth (*x*-axis) crossover in LFP power (*y*-axis). (**f**) Same as e, but for LPC. The LPC depth measure was more correlated with the LFP power reversal than the CSD measure (Pearson's *r* = 0.65 vs. 0.87 CSD vs. LPC).

in V1, where the superficial and deep layers showed stronger spike-count correlations and in the input layers showed weaker correlations (*Hansen et al., 2012*). One might predict that a different pattern of spike-phase coupling would occur in V1, although the contribution of an anesthetized state is unclear.

We find reliable patterns across V4, MT, and PFC, and across two species of monkey performing visually guided tasks. Our findings may be limited to these conditions, as we do not yet know if the same observations hold in cortical areas with non-sensory responses, under conditions of sleep or anesthesia, or in other species. The answer may rely on what generates the observed pattern of spike–LFP phase relationship. While our purpose here is to provide an empirically determined alternative or supplement to CSD analysis, we can speculate about why this observed LPC reversal occurs at the transition from input to deep layers. One possibility stems from the hypothesis that the predominantly radial architecture of cortical fibers between superficial and deep layers forms an electrical dipole that spans across the layers of the cortex (*Buzsáki et al., 2012*). This hypothesis underlies the generation of electrical fields parallel to the dipole measured with EEG recordings at the scalp, and the generation of magnetic fields orthogonal to the dipole measured in MEG recordings. The prediction, therefore, is that the observed spike-phase reversal occurs due to the sources of LFP phase inverting across a dipole generated by the structure of the parallel radial processes of pyramidal cells that arborize in the superficial layers (however see *Riera et al., 2012*). Future experiments dissociating the laminar contribution of cell populations to the LFP at different cortical depths may reveal the degree to which the observed phase reversal is due to the dipole hypothesis or some other circuit mechanism. While the network mechanism responsible for the observed phase reversal remains unclear, our results indicate the reversal is well aligned to the boundary that separates the input and deep layers in multiple cortical areas.

The LPC method for estimating laminar depth described here could be a useful supplement to CSD analysis as it provides a source of cross-validation for potentially noisy or ambiguous CSD patterns. It could also serve as an alternative source of laminar information when CSD analysis is impractical given experimental limitations. The advantages of the LPC method are that it is fast and simple, making it well suited to online depth estimation from unsorted multi-unit spiking activity. LPC can be estimated from ongoing activity in a variety of experimental conditions, cortical areas, and analysis parameters, and may help relate the function of neural populations to the fundamental computation of information processing in the columnar cortical circuit.

## Materials and methods
### Surgical approach
Data from two adult male marmoset monkeys (2 and 3 years of age; *Callithrix jacchus*) and two adult male rhesus macaques (13 and 15 years of age; *Macaca mulatta*) were used in this study. Data from the macaques were previously published in *Franken and Reynolds, 2021*. Macaque surgical procedures have been described before in *Nandy et al., 2017*. Similar techniques were used in the marmoset monkeys. Monkeys were surgically implanted with headposts for head stabilization and eye tracking using cranial screws and dental acrylic or cement. In subsequent surgical sessions a titanium recording chamber was installed in a craniotomy made over area MT or PFC in one marmoset, respectively, or area V4 in both macaques, according to stereotactic coordinates. The dura mater within the chamber was removed, and replaced with a silicone-based optically clear artificial dura, establishing an optical window over the cortex. All surgical procedures were performed with the monkeys under general anesthesia in an aseptic environment in accordance with the recommendations in the Guide for the Care and Use of Laboratory Animals of the National Institutes of Health. All experimental methods were approved by the Institutional Animal Care and Use Committee (IACUC) of the Salk Institute for Biological Studies and conformed with NIH guidelines (protocol 14-00014).

### Electrophysiological recordings
Electrode voltages were recorded from a 32-channel linear silicone electrode array (Atlas Neuroengineering; Leuven, Belgium) connected to an Intan RHD2000 USB interface board (Intan Technologies LLC, Los Angeles, USA) controlled by a Windows computer. The probe was inserted through the artificial dura using a hydraulic microdrive mounted on the chamber using an adjustable *x–y* stage (MO-972A, Narashige, Japan). The probe was lowered until spiking and LFP could be observed on

most of the electrode contacts. Then, the probe was retracted typically by several 100 µm to ease dimpling of the cortex. Data were sampled at 30 kHz from all channels. Neural data were broken into two streams for offline processing of spikes (single- and multi-unit activity) and LFPs. Spike data were high-pass filtered at 500 Hz and candidate spike waveforms were defined as exceeding 4 standard deviations of a sliding 1-s window of ongoing voltage fluctuations. Sorted units were classified as single- or multi-units and single units were validated by the presence of a clear refractory period in the autocorrelogram. LFP data were low-pass filtered at 300 Hz and down-sampled to 1000 Hz.

## Behavioral tasks

Marmosets were trained to enter a custom-built marmoset chair that was placed inside a faraday box with an LCD monitor (ASUS VG248QE) at a distance of 40 cm. The monitor was set to a refresh rate of 100 Hz and gamma corrected with a mean gray luminance of 75 candelas/m$^2$. For the macaques visual stimuli were presented using a LED projector, back-projected on a rear-projection screen that was positioned at a distance of 52 cm from the animal's eyes (PROPixx, VPixx Technologies, Saint-Bruno, Canada). The marmosets and macaques were headfixed by a headpost for all recordings. Eye position was measured with an IScan CCD infrared camera. The MonkeyLogic software package developed in MATLAB (https://www.brown.edu/Research/monkeylogic/; https://monkeylogic.nimh.nih.gov/) (*Asaad et al., 2013*) was used for stimulus presentation, behavioral control, and recording of eye position. Digital and analog signals were coordinated through National Instrument DAQ cards (NI PCI6621) and BNC breakout boxes (NI BNC2090A).

## Inclusions and exclusion criteria

Ongoing data while marmosets and macaques performed a variety of tasks and viewing conditions were used in this study. These conditions included fixation of a fixation point, receptive field mapping, freely viewing natural images, freely viewing in a darkened room, and performing previously described visual tasks (marmoset [*Davis et al., 2020a*], macaque [*Franken and Reynolds, 2021*]). Not all tasks were included in each monkey's experimental battery. Sessions were only included if CSD analysis revealed a recognizable source–sink pattern consistent with reversals across superficial and deep layers. All data unless stated otherwise were collapsed across applicable conditions as the results did not depend on sensory conditions (*Figure 3—figure supplement 4* ). Six recording sessions from V4 in each macaque, 8 recording sessions in marmoset MT, and 14 recording sessions from marmoset PFC were used in the analyses across recordings.

## CSD analysis

A CSD mapping procedure on evoked LFPs was used to estimate the laminar position of recorded channels (*Franken and Reynolds, 2021*; *Nandy et al., 2017*). Mapping stimuli varied across recording locations. For macaque recordings in V4, monkeys maintained fixation while dark gray ring stimuli were flashed (32-ms stimulus duration, 94% luminance contrast, sized and positioned to fall within the cRF of the probe position). For marmoset recordings in MT, monkeys maintained fixation while drifting Gabor stimuli (spatial frequency = 0.5 cycles per degree; temporal frequency = 10 cycles per second, 50% luminance contrast) were presented. For marmoset recordings in PFC, monkeys maintained fixation while full field 100% luminance flashes (background luminance 0.5 candelas/m$^2$; 20-ms flash at 150 candelas/m$^2$) were presented. The CSD was calculated as the second spatial derivative of the stimulus-triggered LFP and visualized as spatial maps after smoothing using bicubic (2D) interpolation (MATLAB function *interp2* with option *cubic*), although the laminar analysis did not critically depend on this particular method of smoothing. Red regions depict current sinks, blue regions depict current sources. We identified the earliest current sink as the input layer. By comparing this position with the range of contacts in the input layer, we could locate channels to superficial, input, or deep layers.

## Generalized Phase

We calculated GP as described previously (*Davis et al., 2022*; *Davis et al., 2020a*). During ongoing spontaneous activity in the visual cortex, LFP signals are composed of fluctuations that vary in amplitude and frequency moment-by-moment, rather than being dominated by a single narrowband oscillation. Instead of applying a narrowband filter that may distort the underlying signal as the frequency content of the signal changes, we use a wideband filter from 5 to 50 Hz (fourth-order Butterworth

forward-reverse filter), which preserves the overall waveform of the LFP as it varies in frequency content from moment to moment. The high-pass serves to eliminate low-frequency fluctuations associated with changes in arousal while the low-pass helps mitigate contamination by spike artifacts in the LFP that would yield spurious spike–LFP coupling relationships. The wideband filtered signal proves a challenge for the standard Hilbert Transform which produces a phase estimate that breaks down quickly for non-narrowband signals, and for this reason it is used strictly in the context of signals pre-treated with a tight narrowband filter (*Le Van Quyen et al., 2001*). The purpose of GP is to mitigate this breakdown of the analytic signal representation for spectrally broad signals by addressing two technical limitations in the Hilbert Transform that occur when applied to broadband signals, described below, and introducing numerical guarantees for the resulting phase representation. GP thus represents an improvement on previous techniques to estimate 'instantaneous phase', as first defined by Denis Gabor (*Gabor, 1946*).

Consider a real-valued signal $x_n \in \mathbf{R}$ for $n \in [1,2,...,N_s]$, where $N_s$ is the number of samples in one recorded trial obtained at a sampling frequency $F_s$. Given $x_n$, its analytic signal representation ($X_n$) is:

$$X_n = x_n + iH[x_n]$$

where $i$ is the imaginary unit and $H[y_n]$ is the Hilbert Transform (HT) of the signal $y_n$. This representation can be obtained by implementing the HT operator as an FIR filter in time domain (*Oppenheim et al., 1999*), or by using a single-sided Fourier transform approach (*Marple, 1999*). The technical limitations in the analytic signal framework occur for two principal reasons. The first limitation is that low-frequency fluctuations effectively shift the representation by a complex constant, which has the critical effect of highly distorting phase angles estimated by the arctangent. As an initial step in the GP representation, then, we filter the signal with a high-pass that excludes low-frequency content. In this work, we also use a low-pass filter at 50 Hz to exclude potential spike waveform artifacts (*Ray and Maunsell, 2011*). After filtering, we then use the single-sided Fourier transform approach (*Marple, 1999*) on the wideband signal and compute phase derivatives as finite differences, which are calculated by multiplications in the complex plane (*Feldman, 2011*; *Muller et al., 2016*; *Muller et al., 2014*). The second limitation is that high-frequency intrusions appear in the analytic signal representation as complex riding cycles (*Feldman, 2011*), which manifest as periods of negative frequencies. As a secondary step in the GP representation, then, we numerically detect these complex riding cycles – namely, $N_c$ points of negative frequency in the phase sequence $Arg[X_n]$ – and utilize shape-preserving piecewise cubic interpolation on the next $2N_c$ points of $Arg[X_n]$ following the detected negative frequency epoch. The resulting representation captures the phase of the dominant fluctuation on the recording electrode at any moment in time, without the distortions due to the large, low-frequency intrusions or the smaller, high-frequency intrusions characteristic of the *1/f*-type fluctuations in cortical LFP (*Linkenkaer-Hansen et al., 2001*; *Milstein et al., 2009*; *Pereda et al., 1998*).

## Laminar analyses

The degree of spike-phase coupling was measured as the mean resultant vector length for the LFP phase distribution collected at the time of observed spikes. This measure was calculated using the circ_r function in the Circular Statistics Toolbox for MATLAB (*Berens, 2009*). The mean phase angle of the spike-phase distribution was calculated using the circ_mean function in the Circular Statistics Toolbox. To generate the cross-channel LPC, the phase of the LFP on each channel was collected for the spike train on each channel. The mean spike-phase angle for each combination of spike and LFP channel was plotted, and the channel that best separated the preferred phase angle was selected by eye as the boundary between the input and deep layers. This process was done blind to the CSD estimate of laminar depth.

For laminar analyses comparing firing rates and LFP power across cortical layers $N = 14$ sessions were selected based on a difference in laminar alignment between CSD and LPC analyses of more than 200 µm. Each channel was aligned to the estimate of the boundary between the input and deep layers in each session based on either the bottom of the earliest current sink in the CSD or the channel preceding the change in the pattern of LPC phase coupling. Mean spike rates at each channel depth were normalized to the mean spike rate on all channels in each session. LFP power in low (8–30 Hz) and high (65–100 Hz) frequency bands were calculated by taking the average

power spectral density in the frequency range of interest at each electrode depth normalized by the average power spectral density in both frequency ranges across all channels. The values were then $z$-scored across channels.

Statistical differences were determined by Wilcoxon signed-rank tests for pairwise differences in the distributions across superficial, input, or deep layers within CSD or LPC conditions, as well as across CSD and LPC conditions. No explicit power analysis was used to determine appropriate sample sizes. Data were curated based on inclusion/exclusion criteria and analyzed across appropriate comparisons (i.e. within/across samples). Differences in circular distributions were computed by Watson–Williams test of homogeneity of means. Linear correlations were calculated using Pearson's $r$. Circular correlations were calculated using the circ_corrcc function for circular–circular correlations and the circ_corrcl function for circular–linear correlations in the MATLAB Circular Statistics Toolbox.

## Acknowledgements

Gatsby Charitable Foundation, the Fiona and Sanjay Jha Chair in Neuroscience, the Swartz Foundation, a NARSAD Young Investigator Grant from the Brain and Behavior Research Foundation, NIH grants R01-EY028723, T32 EY020503-06, P30 EY019005, K99 EY031795, NSF/CIHR through a NeuroNex award (Award No. 2015276), Digital Research Alliance of Canada, and BrainsCAN at Western University through the Canada First Research Excellence Fund (CFREF).

## Additional information

### Funding

| Funder | Grant reference number | Author |
| --- | --- | --- |
| National Eye Institute | R01-EY028723 | John H Reynolds |
| National Eye Institute | T32 EY020503-06 | Zachary W Davis |
| National Eye Institute | P30 EY019005 | John H Reynolds |
| National Eye Institute | K99 EY031795 | Tom P Franken |
| Canadian Institute for Health Research and NSF | NeuroNex Grant No. 2015276 | Lyle Muller |

The funders had no role in study design, data collection, and interpretation, or the decision to submit the work for publication.

### Author contributions

Zachary W Davis, Conceptualization, Resources, Data curation, Software, Formal analysis, Funding acquisition, Validation, Investigation, Visualization, Methodology, Writing – original draft, Project administration, Writing – review and editing; Nicholas M Dotson, Conceptualization, Resources, Data curation, Formal analysis, Investigation, Writing – review and editing; Tom P Franken, Resources, Data curation, Supervision, Funding acquisition, Investigation, Writing – review and editing; Lyle Muller, Conceptualization, Resources, Software, Supervision, Methodology, Writing – review and editing; John H Reynolds, Conceptualization, Resources, Supervision, Funding acquisition, Project administration, Writing – review and editing

### Author ORCIDs

Zachary W Davis http://orcid.org/0000-0003-4440-9011
Nicholas M Dotson http://orcid.org/0000-0002-0885-2182
Tom P Franken http://orcid.org/0000-0001-7160-5152
Lyle Muller http://orcid.org/0000-0001-5165-9890

### Ethics

All surgical procedures were performed with the monkeys under general anesthesia in an aseptic environment in accordance with the recommendations in the Guide for the Care and Use of Laboratory Animals of the National Institutes of Health. All experimental methods were approved by the

Institutional Animal Care and Use Committee (IACUC) of the Salk Institute for Biological Studies and conformed with NIH guidelines (protocol 14-00014).

### Decision letter and Author response
Decision letter https://doi.org/10.7554/eLife.84512.sa1
Author response https://doi.org/10.7554/eLife.84512.sa2

---

## Additional files

### Supplementary files
• MDAR checklist

### Data availability
The source data and code necessary to generate the results in the main figure panels are available at the open-source repository: https://github.com/zwdsalk/LaminarPhaseCoupling (copy archived at *Davis, 2023*). Code for Generalized Phase is available at https://github.com/mullerlab/generalized-phase (*Davis et al., 2020b*). Additional example data are available through Dryad at https://doi.org/10.5061/dryad.3r2280gmm. Correspondence and requests for further materials or assistance in their use should be addressed to JR and ZWD.

The following dataset was generated:

| Author(s) | Year | Dataset title | Dataset URL | Database and Identifier |
| --- | --- | --- | --- | --- |
| Davis ZW, Dotson NM, Franken TP, Muller L, Reynolds JH | 2023 | Laminar Phase Coupling Data | https://doi.org/10.5061/dryad.3r2280gmm | Dryad Digital Repository, 10.5061/dryad.3r2280gmm |

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
