## [Editor Report]

Authors demonstrate powerful methods that can be applied across species to find reliable markers that characterize activity in different cortical layers. Authors provide compelling evidence for these methods that enable systematic comparisons between slow extracellular voltage fluctuations and spiking across cortical columns. The results are timely since linear multielectrode array recordings have become a state-of-the-art technique.

---

## [Decision Letter]

**Decision letter after peer review:**

Thank you for submitting your article "Spike-phase coupling patterns reveal laminar identity in primate cortex" for consideration by *eLife*. Your article has been reviewed by 2 peer reviewers, and the evaluation has been overseen by a Reviewing Editor and John Huguenard as the Senior Editor. The following individual involved in the review of your submission has agreed to reveal their identity: Alexander Thiele (Reviewer #2).

Essential revisions:

1) Authors should define the 'generalized phase' of a 45 Hz-wide signal and the power spectra of the LFP in the analysis.

2) Include Mendoza-Halliday et al. https://www.biorxiv.org/content/10.1101/2022.09.30.510398v1 reference with a discussion of this relevant paper to the present report.

3) Include sample sizes in all Figures.

*Reviewer #2 (Recommendations for the authors):*

The only comment I have is that the authors might want to reference the recent SFN abstract by Andre Bastos about spectral power cross-over in layer IV of different cortical areas.

---

## [Author Response]

Essential revisions:1) Authors should define the 'generalized phase' of a 45 Hz-wide signal and the power spectra of the LFP in the analysis.

We have added a more thorough description of Generalized Phase (GP) to our methods (page 31, line 541) which should convey intuition about the phase of a signal with broad spectral content. We have also included power spectra of the LFP in our revised manuscript that display the broadband, non-stationary, and non-oscillatory spectral content of the LFP in our marmoset and macaque cortical recordings (New Figure 1-supplemental figure 1).

2) Include Mendoza-Halliday et al. https://www.biorxiv.org/content/10.1101/2022.09.30.510398v1 reference with a discussion of this relevant paper to the present report.

We have added a citation to Mendoza-Halliday et al., (2022), in the Discussion section. See text on page 12, Line 278.

3) Include sample sizes in all Figures.

The appropriate sample size used in each figure has been added to each figure legend where it was not already present.

Reviewer #2 (Recommendations for the authors):The only comment I have is that the authors might want to reference the recent SFN abstract by Andre Bastos about spectral power cross-over in layer IV of different cortical areas.

We thank the Reviewer for their helpful suggestion. We believe that the material covered in this SfN abstract appears in the paper Reviewer 1 suggested we cite (Mendoza-Halliday et al., 2022), which we have now included.